# YouTube as a Source of Patient Information Regarding Exercises and Compensated Maneuvers for Dysphagia

**DOI:** 10.3390/healthcare9081084

**Published:** 2021-08-23

**Authors:** Min Cheol Chang, Donghwi Park

**Affiliations:** 1Department of Rehabilitation Medicine, Yeungnam University Hospital, Daegu 42415, Korea; wheel633@ynu.ac.kr; 2Department of Physical Medicine and Rehabilitation, College of Medicine, Ulsan University Hospital, University of Ulsan, Ulsan 44033, Korea

**Keywords:** deglutition, dysphagia, exercise, rehabilitation, YouTube, swallowing difficulty

## Abstract

Objective: Acquiring online health-related information has become increasingly widespread. In this study, we aimed to evaluate the quality of the most-viewed YouTube videos on dysphagia regarding exercises and compensated maneuvers. Method: We searched for the keywords “dysphagia exercise”, “dysphagia rehabilitation”, “dysphagia maneuver”, “dysphagia therapy”, and “dysphagia compensation” on YouTube on 5 February 2021. The educational quality of videos on YouTube was investigated based on the Global Quality Scale (GQS) and categorized into three groups: high-, intermediate-, and low-quality. The modified DISCERN tool was used to evaluate the reliability of the YouTube videos. Video parameters were compared between the groups according to the quality of the videos. Results: Of the 51 videos evaluated, according to the GQS, 54.9% (*n* = 28) were of high-quality, 35.3% (*n* = 18) were of intermediate-quality, and 9.8% (*n* = 5) were of low-quality, respectively. When the video parameters were compared among the groups, there were no significant differences in the number of views, likes, dislikes, or comments per day (*p* > 0.05). However, there was a significant difference in the DISCERN scores between the groups (*p* < 0.001). Conclusion: YouTube can be deemed as a predominant source for high-quality videos on dysphagia exercise and compensated maneuvers. However, YouTube should be accepted as a mixed pool, with high-, intermediate-, and low-quality videos. Therefore, healthcare professionals, such as physicians and therapists, should verify the suitability and quality of the video, and suggest it to the patient, to ensure that the patient obtains the appropriate information.

## 1. Introduction

Dysphagia can be caused by various diseases or problems, including neurological impairment, and physiological and anatomical disorders in any part from the mouth to the esophagus [1,2,3,4]. To treat or improve symptoms of dysphagia, various methods have been reported. Conventional treatment methods such as oropharyngeal exercises and compensated maneuvers, such as the chin-tuck maneuver, are widely used to improve the symptoms of dysphagia [5,6,7,8,9].

The Internet has expanded to become the cardinal means to disseminate information akin to its benefits in the world [10]. Acquiring health information using the Internet has become increasingly common. In a previous study, it was reported that nearly half of adults in the United States (US) obtain health-related information by using the Internet [11]. As a popular video sharing site, YouTube is broadly used around the world to allow users to share and watch videos on the Internet [10]. Because of its free video content and extensive nature, YouTube can be judged as an effective tool for obtaining and spreading health-related information. Therefore, it can also be used as an effective tool for educating patients.

However, there are concerns about the contents and quality of the videos. In particular, considering the nature of YouTube, where anyone can upload videos without verification, and that it can be used for advertising purposes, the quality of the information, content, and accuracy of uploaded videos must be verified. That is, the videos on YouTube can be questioned regarding the risks of providing misleading health-related information and the reliability of the video [12,13]. A previous systemic review that investigated eighteen studies found that YouTube includes misleading and conflicting health-related information, as well as high-quality health-related information [14].

Until now, there have been no studies that have investigated the quality of YouTube videos related to dysphagia exercises and compensated maneuvers. The primary aim of this study is to evaluate the quality of the most viewed English YouTube videos related to exercises and compensated maneuvers for dysphagia. The secondary aim of this study is to investigate the quality of the video sources. The final aim of this study is to analyze the correlations between the number of views, likes, dislikes, and comments among the high-, intermediate-, and low-quality videos on YouTube.

## 2. Material and Methods

In this study, the keywords “dysphagia rehabilitation”, “dysphagia exercise”, “dysphagia therapy”, “dysphagia compensation”, and “dysphagia maneuver” were used when searching for videos on YouTube (http://www.youtube.com, accessed on 5 February 2021). Keywords for the first three pages (60 videos in total) of English-language YouTube videos were individually investigated by two expert physiatrists who have more than 10 years of experience with dysphagia treatment. Previous research showed that a large percentage of YouTube users explore videos on the first three pages of the query results [15,16]. So, we thought that evaluating the first three pages would cover most YouTube users [10]. Then, the videos on YouTube were reviewed according to the number of views. Thus, the most viewed videos are shown at the top. A total of 300 videos were assessed by the two expert physiatrists. Off-topic videos, duplicated videos, videos in a language other than English, and videos with unsuitable sound (e.g., when the quality of sound was so poor that it was difficult to understand.) were excluded from this study. A total of 51 YouTube videos were included, following the exclusion criteria (Figure 1).

### 2.1. Assessment of Quality

The global quality scale (GQS), a tool designed to assess the quality of Internet resources [17], was used by two independent expert physiatrists (MCC, DP) to evaluate the educational quality of the YouTube videos (Appendix A). The GQS is a five-point scale in which the lowest score is 1, and the highest score is 5 [17]. The researchers evaluated the ease of flow, quality, and use of videos using this scale. If a YouTube video obtained a score of 4 or 5 points, it was considered as being of a high quality, if it obtains 3 points it was considered as being of an intermediate quality, and if it obtains 1 or 2 points, it was considered of low quality. If there were disagreements between the two physiatrists’ scores for a video, a consensus was reached through a discussion.

### 2.2. Assessment of Reliability

The modified DISCERN tool, originally created by Charnock et al. [18], was used to evaluate the reliability of the YouTube videos. The modified DISCERN tool includes five questions, and each question is answered as yes or no [18,19]. Each yes answer receives 1 point; the maximum score is 5 (Appendix A) [19].

### 2.3. Video Parameters

The video length, the number of views, the date of upload, the number of likes, the number of dislikes, and the number of comments were examined for each video [10]. The total number of views, likes, dislikes, and comments were divided by the total number of days the video had been on YouTube [10]. Thus, values per day (the number of views, likes, dislikes, and comments per day) were assessed (Table 1).

### 2.4. Sources of the Videos

The sources of the videos were divided into eight categories: (1) Therapists (speech or occupational), (2) physicians or physiatrists, (3) health-related websites, (4) academic, (5) association/professional organization/university, (6) non-physician health personnel, (7) patients, and (8) independent users [10].

### 2.5. Ethics Statement

This study did not involve any human participants or animals. YouTube videos that were accessible to everyone were assessed for this study. Thus, the approval of the ethics committee was not necessary to perform this study. Similar previous studies have followed the same approach. 

### 2.6. Statistical Analysis

To evaluate the correlations among the general features of YouTube videos, a Pearson‘s correlation test was used. To evaluate the difference in the number of views, likes, dislikes and comments per day, and the DISCERN reliability among three groups according to GQS, one-way analysis of variance (ANOVA) with the post hoc Tukey honestly significant difference (HSD) test was used. Using the chi-square test, categorical variables were analyzed. Additionally, using the Kruskal–Wallis test, continuous variables were analyzed. The agreement between the two investigators was evaluated using the kappa coefficient. It was considered significant if the *p*-values were less than 0.05. Statistical analyses were performed using the SPSS version 20.0 for Windows (SPSS Inc., Chicago, IL, USA).

## 3. Results

Of the 300 videos, 121 duplicate videos, 91 off-topic videos, 35 videos in a language other than English, and 2 videos with inappropriate audio were excluded from the study. A total of 51 videos were analyzed. The general characteristics of the videos on YouTube including the video length, number of views, likes, dislikes, and comments are summarized in Table 2. Of the 51 videos evaluated, 54.9% (*n* = 28), 35.3% (*n* = 18), and 9.8% (*n* = 5) were of high-, intermediate-, and low-quality, respectively, according to the GQS. According to sources, when the distribution of high-quality videos was analyzed, it was discovered that 70.0% (*n* = 14) of the YouTube videos were made by universities/professional organizations/associations, 57.1% (*n* = 8) of the YouTube videos were made by a therapist, 50.0% (*n* = 2) of the YouTube videos were made by physicians or physiatrists, 50.0% (*n* =1) of the videos were made by non-physician health personnel, and 30.0% (*n* = 3) of the videos were made by a health-related website (Table 2). Among the general features of YouTube videos, the number of views showed statistically significant correlations with the number of likes, dislikes, and comments (Table 3). Moreover, number of comments showed statistically significant correlations with the number of likes, dislikes, and views (Table 3).

When the parameters of the videos were compared among the high-, intermediate-, and low-quality groups, there were no significant differences found in the number of views, likes, and comments per day (*p* > 0.05). However, there was a significant difference in the DISCERN scores between the groups (*p* < 0.001) (Table 4). In the post hoc analysis, the DISCERN scores between the low- and intermediate-quality and between the low- and high-quality showed significant differences (*p* < 0.05) (Table 4). The kappa score, which shows the inter-reviewer agreement of this study, was 0.891.

## 4. Discussion

As one of the most preferred video-sharing websites, YouTube has many videos about the diagnosis, treatment, etiopathogenesis, and prevention of various diseases [20]. It provides free video content to website users, but lacks control mechanisms for the quality and accuracy of the videos. Additionally, anyone with a YouTube account can upload videos. So, this may cause the spread of poor quality, incorrect, or biased information. Keelan et al. [21], who was the first to perform research on the quality of YouTube videos, evaluated the quality of immunization-related YouTube videos. Consequently, other investigators assessed the quality of YouTube video associated with various diseases [10,16,22,23,24,25,26].

However, the quality of YouTube videos about exercises or compensated maneuvers for dysphagia has not yet been investigated. Therefore, in this study, YouTube videos about exercises or compensated maneuvers for dysphagia were evaluated by asking “What is the quality the information offered by English language YouTube videos about exercises or compensated maneuvers for dysphagia?”, “Is there any difference in terms of the number of views, likes, dislikes, and comments according to the quality of YouTube videos?”, “Which resources have uploaded high-quality YouTube videos?”, and “Which resources have uploaded the most YouTube videos?”

According to the GQS, 9.8% (*n* = 5), 35.3% (*n* = 18), and 54.9% (*n* = 28) of the videos evaluated were of and low-quality, intermediate-quality, and high-quality, respectively. In previous studies, however, different useful or high-quality YouTube video ratios conducted on various diseases have been reported. 

Similar to our results, Singh et al. [22], Garg et al. [24], and Tolu et al. [23] reported that about half of YouTube videos were useful based on the assessment of quality. However, Nason et al. [25], Sahin et al. [26] and Rittberg et al. [15] reported that the proportions of the most useful or high-quality YouTube video were 18.4%, 2.0%, and 19.6%, respectively. There may be many reasons for the conflicting results regarding the quality of YouTube videos. First, the topics of videos on YouTube evaluated in the previous research were different pathologic conditions or diseases, such as the methotrexate self-injection technique, rheumatoid arthritis, dialysis, bariatric surgery, subcutaneous anti-tumor necrosis factor agent injections, and urethral catheterization. Therefore, depending on the topics of the videos, the quality may vary significantly. In the results of this study, YouTube videos about exercises and compensated maneuvers for dysphagia are of relatively high-quality. Perhaps dysphagia is an area that the public is less informed about, and therefore cannot upload videos about it. Most of the YouTube videos about dysphagia had been uploaded by physicians, therapists, and medical institutions. The subject nature of dysphagia makes it difficult to upload misleading health information, and videos about exercise and compensated maneuvers for dysphagia seemed to be relatively accurate. However, regarding more general topics, unlike dysphagia, it is thought that the opposite result may be possible. Therefore, patients should not trust YouTube videos unconditionally, and it is important to check the information with therapists or physicians at least once.

Second, the evaluation of the quality of YouTube videos is subjective, and it can be different for each study because of the absence of objective criteria. Moreover, the number of videos assessed in the studies is different. 

Our study has shown the importance of clarifying the sources when using YouTube as a source of health-related information. When the quality of the videos is assessed according to sources, the primary sources of the highest quality videos are university/professional organization/association, followed by therapists, physicians or physiatrists, and non-physician health personnel. Intermediate- and low-quality videos are sourced from health-related websites, therapists, and patients. Consistent with our results, in previous studies, high-quality videos were sourced from healthcare professionals or organizations, and the primary sources of low-quality videos were medical advertisements, for-profit organizations, and independent users.

The number of views is the most significant marker of the popularity of YouTube videos. Persons who watch videos on YouTube can click the “like” or “dislike” button according to their ratings and comment under the videos. However, between the quality of the videos and the number of views, likes, dislikes, and comments, there were no significant correlations in this study. However, our results show that high-quality videos are also more reliable. 

This study has a few limitations. First, we assessed the quality of the video according to the GQS, which is subjective. Moreover, we only investigated the videos from a single point of view. This research method has limitations, since videos are continuously uploaded to YouTube. Second, we only searched for YouTube videos in the English language. Therefore, previous Internet activity and geographic location may affect the results of the search. Furthermore, studies with multiple different languages and points of view are warranted to overcome these limitations. Third, we used the clinical term “Dysphagia” in the search on YouTube. However, given that patients are more likely to look for “swallowing exercise” instead of “dysphagia” in YouTube, further studies with more colloquial terms are necessary. Fourth, in the assessment of GQS of this study, a consensus was reached through a discussion when there were discrepancies between the two physiatrists’ scores for a video. It is believed that a more objective result could have been drawn if the conclusion was drawn through a third specialist. Lastly, YouTube logarithms could take information from cookies and other sources that could influence the searches. Therefore, our YouTube search results may not be consistent for everyone. In the future, further research is needed to overcome this by using a method that can reduce the variables caused by YouTube’s logarithms, such as synthesizing all search results from various people in various countries.

## 5. Conclusions

In this study, we evaluated the quality of the most viewed YouTube videos on dysphagia regarding exercise and compensated maneuver, investigated the quality of the video sources, and analyzed the correlations among the quality, reliability, and parameters of video parameters. YouTube can be deemed as a predominant source for high-quality videos on dysphagia exercise and compensated maneuvers. However, YouTube should be accepted as a mixed pool, with high-, intermediate-, and low-quality videos. Therefore, healthcare professionals, such as physicians and therapists, should verify the suitability and quality of the video, and suggest it to the patient, to ensure that the patient obtains the appropriate information. 

## Figures and Tables

**Figure 1 healthcare-09-01084-f001:**
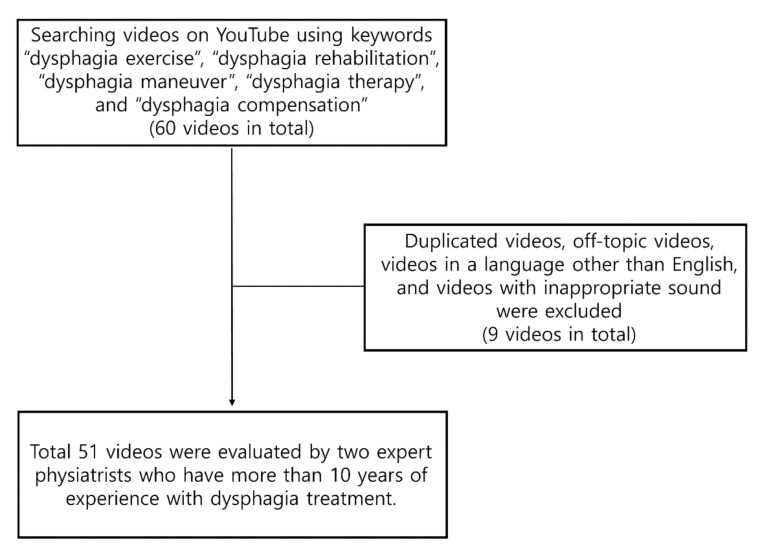
The flowchart of this study.

**Table 1 healthcare-09-01084-t001:** General features of the videos.

Video Features	Mean ± SD (Min–Max)
Duration (days)	394.16 ± 893.12 (36–6000)
Number of views	4,795,859 ± 10,860 (20–699,485)
Number of likes	416.67 ± 1058.95 (0–6800)
Number of dislikes	18.49 ± 44.91 (0–281)
Number of comments	40.06 ± 90.45 (0–375)

SD; standard deviations.

**Table 2 healthcare-09-01084-t002:** Pearson‘s correlations among the general features of the videos.

	Duration (s)	Number of Views	Number of Likes	Number of Dislikes	Number of Comments
Duration (s)	1				
Number of views	−0.54 (0.707)	1			
Number of likes	−0.35 (0.809)	**0.950 *** (<0.001)	1		
Number of dislikes	−0.57 (0.691)	**0.987 *** (<0.001)	**0.938 *** (<0.001)	1	
Number of comments	−0.26 (0.855)	**0.583 *** (<0.001)	**0.621 *** (<0.001)	**0.655 *** (<0.001)	1

* *p* < 0.05 (bold), Spearman’s rank correlation coefficient (*p*-value).

**Table 3 healthcare-09-01084-t003:** Categorization of the videos according to sources, *n* (%).

Source	Low Quality	Intermediate Quality	High Quality	Total
Therapists	2 (14.3)	4 (28.6)	8 (57.1)	14
Physician or physiatrists	0 (0.0)	2 (50.0)	2 (50.0)	4
Health-related website	0 (0.0)	7 (70.0)	3 (30.0)	10
Academic	0 (0.0)	0 (0.0)	0 (0.0)	0
University/professional organization/association	2 (10.0)	4 (20.0)	14 (70.0)	20
Nonphysician health personnel	0 (0.0)	1 (50.0)	1 (50.0)	2
Patient	1 (100.0)	0 (0.0)	0 (0.0)	1
Independent user	0 (0.0)	0 (0.0)	0 (0.0)	0

**Table 4 healthcare-09-01084-t004:** Comparison of the video parameters between the low-quality, intermediate-quality and high-quality groups.

Video Quality	DISCERN Score Mean ± SD (Min–Max)	View per Day Mean ± SD (Min–Max)	Like per Day Mean ± SD (Min–Max)	Dislike per Day Mean ± SD (Min–Max)	Comment per Day Mean ± SD (Min–Max)
Low	1.6 ± 1.14 (0–3)	97.05 ± 148.59 (2.17–359.86)	0.84 ± 1.31 (0.01–3.14)	0.09 ± 0.12 (0.0–0.25)	0.56 ± 0.85 (0.0–1.54)
Intermediate	3.0 ± 0.49 (2–4)	137.30 ± 253.87 (0.48–988.98)	0.80 ± 1.38 (0.01–4.27)	0.12 ± 0.17 (0.0–0.51)	0.46 ± 0.64 (0.0–1.77)
High	3.39 ± 0.74 (2–5)	286.85 ± 600.76 (0.0–2562.22)	2.45 ± 5.16 (0.0–24.91)	0.13 ± 0.26 (0.0–1.03)	0.17 ± 0.32 (0.01–1.11)
*p*-value	<0.001	0.497	0.397	0.947	0.244

SD; standard deviations.

## Data Availability

The data presented in this study is available on request from the corresponding author.

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
