# Peer review of "YouTube as a Source of Patient Information Regarding Exercises and Compensated Maneuvers for Dysphagia"

_healthcare, 2021, doi:10.3390/healthcare9081084_

Round 1

Reviewer 1 Report

First of all, I Want to note that it has been a pleasure review your manuscript. 
I think this is an interesting topic forma clinicians and  patients who mánager this prevalent condición.

In the discussion and conclusions, you should explain how to improve the quality of videos for youtube patients.

Furthermore, it does not explain the limitations of the study.

Author Response

Answers to Reviewer`s Response

Reviewer 1

First of all, I Want to note that it has been a pleasure review your manuscript.

I think this is an interesting topic forma clinicians and patients who mánager this prevalent condición.

In the discussion and conclusions, you should explain how to improve the quality of videos for youtube patients.

Furthermore, it does not explain the limitations of the study.

Answer: We appreciate your valuable comment. Following your comment, we further reinforced the content of the limitations part of this study. YouTube should be accepted as a mixed pool with high-, intermediate-, and low-quality videos. Therefore, to improve the quality of videos, healthcare professionals, such as physicians and therapists, should verify the suitability and quality of the video, and suggest it to the patient, to ensure that the patient gets the appropriate information.

Reviewer 2 Report

The present manuscript is an interesting approach to the one of the new frontiers of health. I agree the authors about the need of assess whether the health information from platforms as YouTube is of quality or not. However, it must be considered that YouTube engine of search do not necessary works as sciences e-databases search engines, as it presents its own private logarithms that could influence the data obtained. In this line, methodologic issues must be solved:

-ABSTRACT, lines 19-20 (Results): please rewrite the sentence as it is not clear enough.

-The authors used a clinical term as “Dysphagia” in the search on YouTube while keywords include some more colloquial terms as “swallowing difficulty” or “deglution”. Authors must consider that is more likely that patients look for “swallowing exercise” instead of “dysphagia” in YouTube. Please address this as limitation on explain why each term was used.

INTRODUCTION

-Lines 54-55: existing studies must be referred or cited.

MATERIAL AND METHODS

-YouTube logarithms could take information from cookies and other sources that could influence in the searches. How did authors take care of this? Even the country from where the search is made could influence.

-Lines 65: used the same size of letter for the link.

-Lines 66: why only three pages of results were checked? You said that previous studies (in plural) used only the three first pages, but only cited one work, this is inconsistent.

-Line 74: please explain what was considered as “inappropriate sound”.

-Line 75: Maybe a flow diagram would help readers to understand the selection process, please consider it in Results section (here in Methods is not appropriate to mention how many studies were finally selected, as this is a result).

-Line 84: the most common way of solving any discrepancies in these types of reviews (and the recommended by scientific guidelines) is by asking a third person, aspect that should be mentioned in limitations.

-Table 1: points 1-5 should start at the same column, no one of them do.

-Table 2: same problem than table 1. Moreover, the items of this questionnaires/tools should not be part of the manuscript, maybe part of supplementary materials to be published online. Have the authors permission from original authors to reproduce these tools in open access?

RESULTS

-Table 4: it would be interesting to consider if there is a correlation between the quality of the videos and the source.

-Table 5: please check “quality” in the first column and use only one line of text for “intermediate”. Define “SD” in footnotes of the table.

DISCUSSION

Please consider to use more updated references, for example 10.2196/mededu.8527 (2018).

Lines 173-174: you refer three authors, but four percentages, which does not fit.

Line 209: did you mean that high-quality videos were better evaluated by the public?

CONCLUSIONS

Them must answer succinctly the Objectives of the study:

-evaluate the quality of the most viewed English YouTube videos

-investigate the quality of the video sources

-analyze the correlations between the number of views, likes, dislikes...

Please modify the conclusions according to the objectives proposed.

Author Response

Answers to Reviewer`s Response

Reviewer 2

The present manuscript is an interesting approach to the one of the new frontiers of health. I agree the authors about the need of assess whether the health information from platforms as YouTube is of quality or not. However, it must be considered that YouTube engine of search do not necessary works as sciences e-databases search engines, as it presents its own private logarithms that could influence the data obtained. In this line, methodologic issues must be solved:

-ABSTRACT, lines 19-20 (Results): please rewrite the sentence as it is not clear enough.

Answer: We appreciate your valuable comment. Following your comment, we have modified it as follows.

“Of the 51 videos evaluated, according to the GQS, 54.9% (n = 28) were of high-quality, 35.3% (n = 18) were of intermediate-quality, and 9.8% (n = 5) were of low-quality, respectively.”

-The authors used a clinical term as “Dysphagia” in the search on YouTube while keywords include some more colloquial terms as “swallowing difficulty” or “deglution”. Authors must consider that is more likely that patients look for “swallowing exercise” instead of “dysphagia” in YouTube. Please address this as limitation on explain why each term was used.

Answer: We appreciate your valuable comment. We totally agree with your comment. Following your comment, we have added it in limitation part.

“Third, we used a clinical term as “Dysphagia” in the search on YouTube. However, given that patients are more likely to look for “swallowing exercise” instead of “dysphagia” in YouTube, further studies with more colloquial terms are necessary.”

INTRODUCTION

-Lines 54-55: existing studies must be referred or cited.

Answer: We appreciate your comment. Actually, there was a mistake. We have modified it as follows;

“Until now, there was no study that investigated the quality of YouTube videos related to dysphagia exercises and compensated maneuver.”

MATERIAL AND METHODS

-YouTube logarithms could take information from cookies and other sources that could influence in the searches. How did authors take care of this? Even the country from where the search is made could influence.

Answer: We appreciated your valuable comment. We totally agree with your comment. We added it in the limitation as follows.

“Lastly, YouTube logarithms could take information from cookies and other sources that could influence in the searches. Therefore, our YouTube search results may not be consistent for everyone. In the future, further research is needed to overcome this by using a method that can reduce the variables caused by YouTube's logarithms, such as synthesizing all search results from various people in various countries.”

-Lines 65: used the same size of letter for the link.

Answer: We modified the link of YouTube.

-Lines 66: why only three pages of results were checked? You said that previous studies (in plural) used only the three first pages, but only cited one work, this is inconsistent.

Answer: We appreciate your valuable comment. We analyzed only the first three pages based on the results that most people watch the majority on the first three pages (60 videos) of YouTube searches in previous studies. Moreover, we added an additional reference in the manuscript.

-Line 74: please explain what was considered as “inappropriate sound”.

Answer: We appreciate your valuable comment. Following your comment, we have modified it as follows;

“Duplicated videos, off-topic videos, videos in a language other than English, and videos with inappropriate sound (ex. when the sound quality is so poor that it is difficult to understand what is being explained.) were excluded from this study.”

-Line 75: Maybe a flow diagram would help readers to understand the selection process, please consider it in Results section (here in Methods is not appropriate to mention how many studies were finally selected, as this is a result).

Answer: We appreciate your valuable comment. Following your comment, we have added the flowchart as figure 1.

-Line 84: the most common way of solving any discrepancies in these types of reviews (and the recommended by scientific guidelines) is by asking a third person, aspect that should be mentioned in limitations.

Answer: We appreciate your valuable comment. We agree with the reviewer's opinion to some extent. However, since two specialists with more than 10 years of experience in treating swallowing disorders came up with a result that both of them can agree with, it is unlikely that it would have had a significant impact on the result. However, we have added it in the limitation part of our study.

“Fourth, in the assessment of GQS of this study, a consensus was reached through a discussion when there were discrepancies between the two physiatrists’ scores for a video. It is believed that a more objective result could have been drawn if the conclusion was drawn through a third specialist.”

-Table 1: points 1-5 should start at the same column, no one of them do.

Answer: We appreciate your valuable comment. We modified it.

-Table 2: same problem than table 1. Moreover, the items of this questionnaires/tools should not be part of the manuscript, maybe part of supplementary materials to be published online. Have the authors permission from original authors to reproduce these tools in open access?

Answer: We appreciate your valuable comment. Following your comment, we modified it as a supplementary files. We also added a reference in the open access journal which use DISCERN tool.

RESULTS

-Table 4: it would be interesting to consider if there is a correlation between the quality of the videos and the source.

Answer: We appreciate your valuable comment. We totally agree with the reviewer's opinion. Following your comment, we have results of correlations among the general features of YouTube videos as Table 3.

-Table 5: please check “quality” in the first column and use only one line of text for “intermediate”. Define “SD” in footnotes of the table.

Answer: We appreciate your valuable comment. Following your comment, we have modified it.

DISCUSSION

Please consider to use more updated references, for example 10.2196/mededu.8527 (2018).

Answer: We appreciate your valuable comment. Following your comment, we have added it as a reference.

Lines 173-174: you refer three authors, but four percentages, which does not fit.

Answer: We appreciate your valuable comment. There was a mistake. We modified it.

“However, Nason et al., 24 Sahin et al.,25 and Rittberg et al.14 have reported the most useful or high-quality YouTube video ratios were 18.4%, 2.0%, and 19.6%, respectively.”

Line 209: did you mean that high-quality videos were better evaluated by the public?

Answer: Both GQS score indicating video quality and DISCERN tool indicating reliability were analyzed by specialists. However, the number of comments, likes, and dislikes, which are indirect public judgment indicators, had no correlation with quality. So, according to specialist`s analysis, it means that high-quality videos have a high reliability score.

CONCLUSIONS

Them must answer succinctly the Objectives of the study:

-evaluate the quality of the most viewed English YouTube videos

-investigate the quality of the video sources

-analyze the correlations between the number of views, likes, dislikes...

Please modify the conclusions according to the objectives proposed.

Answer: We appreciate your valuable comment. Following your comment, we have modified it as follows’

“In this study, we evaluated the quality of the most viewed YouTube videos on dysphagia on exercise and compensated maneuver, investigated the quality of the video sources, and analyzed the correlations among the quality, reliability, and parameters of video parameters. YouTube can be deemed as a predominant source for high-quality videos on dysphagia exercise and compensated maneuvers. However, YouTube should be accepted as a mixed pool with high-, intermediate-, and low-quality videos. Therefore, healthcare professionals, such as physicians and therapists, should verify the suitability and quality of the video, and suggest it to the patient, to ensure that the patient gets the appropriate information”

Round 2

Reviewer 2 Report

Thank you for your effort. The study still present some methodological flaws due to the complexity of Youtube Search engine, but I consider they are very difficult to solve and all of them have been named at limitations.